# A Patch-Based CNN Built on the VGG-16 Architecture for Real-Time Facial Liveness Detection

Dewan Ahmed Muhtasim ⬛, Monirul Islam Pavel ⬛ and Siok Yee Tan *⬛





Center for Artificial Intelligence Technology, Faculty of Information Science and Technology,
Universiti Kebangsaan Malaysia, Bangi 43600, Malaysia
* Correspondence: esther@ukm.edu.my

**Abstract:** Facial recognition is a prevalent method for biometric authentication that is utilized in a variety of software applications. This technique is susceptible to spoofing attacks, in which an imposter gains access to a system by presenting the image of a legitimate user to the sensor, hence increasing the risks to social security. Consequently, facial liveness detection has become an essential step in the authentication process prior to granting access to users. In this study, we developed a patch-based convolutional neural network (CNN) with a deep component for facial liveness detection for security enhancement, which was based on the VGG-16 architecture. The approach was tested using two datasets: REPLAY-ATTACK and CASIA-FASD. According to the results, our approach produced the best results for the CASIA-FASD dataset, with reduced HTER and EER scores of 0.71% and 0.67%, respectively. The proposed approach also produced consistent results for the REPLAY-ATTACK dataset while maintaining balanced and low HTER and EER values of 1.52% and 0.30%, respectively. By adopting the suggested enhanced liveness detection, architecture that is based on artificial intelligence could make current biometric-based security systems more secure and sustainable while also reducing the risks to social security.

**Keywords:** biometric; liveness detection; social security; artificial intelligence; CNN; VGG-16; LSTM

## 1. Introduction

Biometric systems have been utilized in various security applications in recent years due to ongoing research into their implementation [1,2]. Facial recognition-based liveness detection is one of the major branches of biometric technology that have been effectively applied in e-commerce, device security and organizational attendance, as well as for ensuring top-notch security, especially in the era of the IR 4.0. The core role of liveness detection is to verify whether the source of a biometric sample is a live human being or a fake representation. This process provides more safety and improvements to traditional facial recognition-based security systems [3], which use a person's unique biometric information, such as their face, to allow that individual to access specific systems or data. However, one of the primary impediments to biometric identification systems is the risk of spoofing attacks [4]. A facial spoofing attack is an attempt by an unauthorized person to circumvent the facial authentication protocol and the facial verification process by employing deception techniques, such as identity forgery [5]. A printed image of an authorized face or a recorded video from a display may provide sufficient unique data to deceive the system [6,7]. As a result, the resiliency of these security systems can be diminished.

A multitude of applications use facial biometric authentication, such as automated teller machines (ATMs), smart security systems and other similar systems [8,9]. Advanced artificial intelligence models increase the data processing capability of biometric technology, which results in more effective biometric security systems [10]. Nevertheless, spoofing attacks are a common form of attack that reduce the effectiveness of biometric authentication systems [11]. A facial spoofing attack is an attempt by an illegal user to circumvent a facial

authentication system and facial verification system using deception methods, such as counterfeiting the identity of an authorized user [5]. Therefore, the implementation of liveness detection systems that are powered by artificial intelligence has the potential to make existing biometric-based security systems more sustainable and secure while also providing better social security.

Recently, numerous effective facial anti-spoofing methods have been developed [4–8]. They can be generally divided into fixed feature-based facial anti-spoofing algorithms [12] and automated learnable feature-based facial anti-spoofing algorithms [13]. Facial anti-spoofing techniques utilize hand-crafted features from actual and counterfeit faces to detect spoofing. The features of actual and false faces are determined before training facial anti-spoofing algorithms. Motion, texture, image quality, 3D shape and multi-spectral reflectance are examples of fixed feature-based algorithms. Automated learnable feature-based facial anti-spoofing methods distinguish between legitimate and fake faces using deep learning approaches, such as convolutional neural networks (CNNs). CNNs learn the properties of real and fake faces during training. By transmitting raw pixels through hidden layers, CNNs translate the raw pixels of images into probabilities. The number of hidden layers determines the depth of a CNN [14,15]. Although deep CNNs are the optimal solution for most applications, their usage in facial anti-spoofing applications has been restricted due to a lack of training data.

This paper proposes a patch-based CNN that was built using the VGG-16 architecture with a deep aspect for identifying liveness in a network. In the proposed architecture, input images are sent sequentially to the CNN, which acts as the front-end of the architecture once the patches have been created. In the next step, the CNN output is passed to an LSTM, which detects temporal information in the sequence and categorizes the dense layer in the neural network output as live or fake. The contributions of this research can be summarized as follows:

- The introduction of a facial recognition-based approach for biometric authentication systems to detect liveness and improve authentication system efficiency;
- A demonstration of a patch-based CNN–LSTM model that can overcome the overfitting and lower accuracy issues of facial anti-spoofing methods using two major datasets.

## 2. Literature Review

### 2.1. Feature-Based Facial Anti-Spoofing Approaches

Liu et al. proposed an improved local binary pattern for face maps that could be used as a classification feature [16]. When these characteristics were put into a support vector machine (SVM) classifier, the face maps could be classified as genuine or false. Tan et al. proposed a technique for formulating the anti-spoofing objective as a binary classification problem [17]. They implemented Difference of Gaussian (DoG) filtering to eliminate noise from 2D Fourier spectra and then extended a sparse logistic regression classifier both nonlinearly and spatially for the classification. Matta et al. utilized multiscale local binary patterns to examine the texture of facial images [18]. Afterward, the macrotexture patterns were encoded into an improved feature histogram, which was then input into an SVM classifier.

Parveen et al. proposed a texture descriptor that they described as the dynamic local ternary pattern, in which the textural features of skin were examined using an adaptive threshold configuration and the classification was performed using a support vector machine with a linear kernel [19]. Das et al. proposed a method that was based on a frequency and texture analysis to distinguish between real and artificial faces [20]. The frequency evaluation was accomplished by Fourier transforming the images into the frequency domain and then calculating the frequency descriptor to detect dynamic variations in the faces. LBP was employed to analyze the texture and an SVM classifier with a radial basis function kernel was used to classify the generated feature vectors.

The method that was proposed by Luan et al. involved three characteristics: blurriness, the specular reflection ratio and color channel distribution characteristics [21]. The images

were classified using an SVM, according to these three characteristics. Chan et al. developed a technique that utilized flash to defend against 2D spoofing attacks [22]. This technique captured two images per individual: one with flash and the other without. Three additional descriptors were employed to extract the textural information of the faces: a descriptor that was based on uniform LBP and the standard deviation and mean of the grayscale difference between the two recorded images of each person. The difference between the pictures with and without flash, as assessed by the four descriptors, was used to classify the images. Kim et al. proposed a method that defined the differences between the surface characteristics of live and fake faces by calculating the diffusion speeds and extracting anti-spoofing characteristics that were based on the local patterns of the diffusion speeds [23]. These characteristics were input into a linear SVM classifier to assess the liveness of the images. Yeh et al. developed a method that utilized digital focus properties with different depths of field to accomplish liveness detection [24]. The nose and the lower right portion of the cheek were examined for preprocessing. Due to the impact of the depth of field, the degree of blurriness differed between real and false images. The k-nearest neighbor method was used to classify the results. Although hand-crafted feature extraction was also utilized in [12,13], we adopted a patch-based CNN that was built on VGG-16 architecture to conduct feature extraction automatically, which removed the need for hand-crafted feature extraction.

### 2.2. Deep Learning-Based Facial Anti-Spoofing Approaches

Deep CNN models have been employed in recent facial liveness identification studies because they provide more accurate liveness detection than the previously presented strategies [7,8,14,15]. For facial anti-spoofing detection, Atoum et al. suggested a two-stream CNN-based method that included a patch-based CNN and a depth-based CNN [25]. While the first CNN extracted local features from face image patches, the second extracted depth features by computing the depth from the entire image and then using an SVM for feature extraction. This two-stream network could be trained end-to-end to discriminate between real and fake faces based on their rich appearance attributes by including these parameters. Rehman et al. proposed a technique that concentrated on data randomization in mini-batches to train deep CNNs for liveness detection [26].

The method that was proposed by Alotaibi et al. used a mix of input facial diffusion accompanied by a three-layer CNN architecture [27]. For facial liveness detection, Alotaibi et al. suggested an approach that employed nonlinear diffusion, accompanied by a tailored deep convolutional network [28]. By rapidly diffusing the input image, nonlinear diffusion aided in differentiating between fake and genuine images. As a result, the edges of flat images faded away while the edges of genuine faces stayed visible. Furthermore, to extract the most significant properties for classification, a customized deep convolutional neural network was suggested. Koshy et al. proposed a method that combined nonlinear diffusion with three architectures: CNN-5, ResNet50 and Inception v4. They found that the Inception v4 architecture was the best [29]. Jourabloo et al. addressed the facial anti-spoofing problem as an image denoising problem, which resulted in the development of an anti-spoofing method that was based on CNNs to achieve the facial anti-spoofing objective [30]. In the first layer of a Lenet-5-based neural network model, De Souza et al. applied the LBP descriptor, which improved the accuracy of facial spoofing detection [31]. An improved version of LBPnet, which is called n-LBPnet, was suggested to achieve higher accuracy for real-time facial spoofing detection by integrating the local response normalization (LRN) step into the second layer of the network.

Xu et al. demonstrated that facial anti-spoofing in videos could be achieved using a deep architecture that integrated LSTM and a CNN [32]. The LSTM obtained the temporal correlations in the input sequences while the CNN retrieved the local and dense features. Tu et al. proposed that facial spoofing detection in video sequences could be achieved using a CNN–LSTM network, which concentrated on motion cues throughout video frames [33]. To enhance the facial emotions of the humans in the videos, they used Eulerian

motion magnification. Highly discriminative features were extracted from the video frames using a CNN and LSTM, which were also used to capture the temporal dynamics from the videos. Recently, Khade et al. proposed an iris liveness detection technique that used several deep convolutional networks [34]. Densenet121, Inceptionv3, VGG-16, EfficientNetB7 and Resnet50 were employed in that study to identify iris liveness using transfer learning approaches. The limited dataset necessitated the use of a transfer learning approach to prevent overfitting. As stated above, numerous well-referenced CNN models have demonstrated the ability to distinguish between real and fake faces. Therefore, this paper presents a patch-based CNN architecture for training complicated and differentiating features to improve the security of present facial recognition-based biometric authentication systems against printed images and replay attacks.

## 3. Architecture of the Proposed System

### 3.1. Liveness Detection

A patch-based CNN that was built on the VGG-16 architecture with a deep aspect was proposed for liveness detection to improve security. After the patches are constructed, the input images are transmitted sequentially to the CNN, which serves as the front-end of the architecture. The CNN output is then passed to an LSTM, which identifies temporal features in the sequence and determines whether the dense layer in the neural network output is real or fake. The workflow of the proposed method is shown Figure 1.

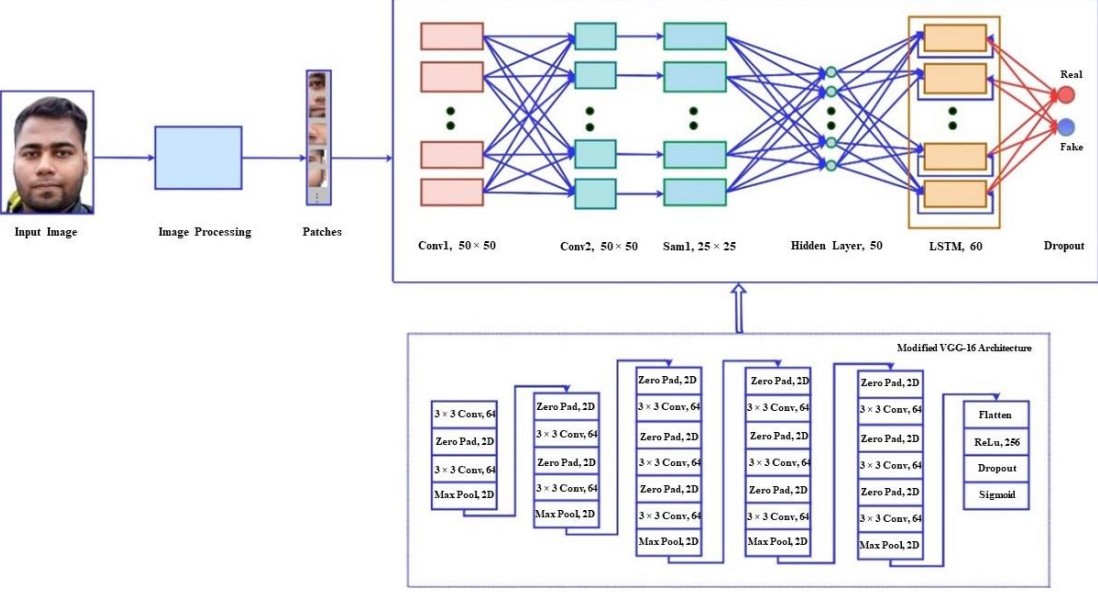

**Figure 1.** The workflow of the patch-based CNN–LSTM architecture with the modified VGG-16.

### 3.2. Patch-Based CNN

Before applying the patch-based approach [23,24,35] for real-time testing purposes, an image processing algorithm (LBPH) was adopted using OpenCV [36], which allowed facial boundaries to be detected within images. Further, there were several motivations for the proposed CNN to use patches instead of the whole face. Firstly, the number of CNN learning samples needed to be increased due to the limited number of samples that were available for training in all of the accessible anti-spoofing datasets. Although hundreds of faces could be taken from individual videos by cropping faces frame by frame, overfitting posed a huge problem while training the CNN because of the significant similarities between the images.

Secondly, classic CNNs need to redimension faces when employing full facial photos as inputs because of the various resolutions of the photos. These shifts in size could lead to a drop in discrimination between images. In contrast, by adopting local patches, the native

resolution of the original images can be maintained and the discriminatory capability of the system can be preserved. To solve this, a patch-based CNN was deployed in which each frame was converted into patches that were then classified separately. As the spoof-specific discriminatory information was present in the whole facial region, patch-level inputs were used to enable the CNN to detect this information irrespective of the patch position, despite this being a more complicated process than using the entire facial image.

Furthermore, the input features were selected by converting HSV color space to obtain the discriminative descriptors as the scope of anti-spoofing methods that use RGB images is limited [25] and color space can also be utilized for chrominance and luminance information. Then, pixel-wise LBP maps were randomly extracted for spatial texture information rather than using them as traditional histogram descriptors [37–39]. Using a pre-trained Haar cascade model for front facial detection, the maps were transformed into feature representations, as well as fixed sized patches, for processing using the VGG-16-based CNN–LSTM model.

### 3.3. Modified VGG-16 Architecture

In this study, VGG-16 was adopted to structure the CNN as it can generalize data better and produce less overfitting with 138 million parameters, as well as being more trainable and variable and performing betters than other CNN architectures [40–42]. Though other CNN architectures are deeper than VGG-16, it is significantly smaller because it uses global average pooling rather than only using FCN.

VGG-16 is a 2D CNN architecture that uses $224 \times 224$ images as the input. This architecture contains 16 layers, of which 13 are convolutional layers and 3 are fully connected layers. It has 64 filters in the first block, 128 filters in the second block, 256 filters in the third block and 512 filters in the fourth and fifth blocks that are available for each convolution [43].

In our model, the first layer was a convolutional layer (Conv1), which comprised 12 characteristic maps with a size of $56 \times 56$, in which each unit of the maps was the result of a convolution of the local reception field using a $3 \times 3$ kernel of the input image. The second layer comprised 18 maps that were $50 \times 50$ and the function maps in Conv1 transformed the feature maps into $7 \times 7$ kernels. A subsampling layer (S2) followed Conv2 and had 18 characteristic maps, which measured $25 \times 25$ and were constructed using a pooling of size (2, 2) to match the Conv2 characteristic maps, thereby halving the resolution of the characteristic maps in Conv2. The following layer consisted of a completely linked 50-neuron hidden layer.

The second layer (Conv2) also consisted of a total of 18 feature maps that were $50 \times 50$, which were obtained from the convolution of the feature maps with $7 \times 7$ kernels in Conv1. The following layer consisted of Conv2 with a subsampling layer (S1) that had 18 features and was $25 \times 25$, which was obtained from the pooling layer with a window size of $2 \times 2$, which reduced the resolution of the feature maps in Conv2 by half.

The third layer contained 50 layers of neurons that were based in a fully connected hidden layer and two fully connected output layers. A dropout probability of 0.25 was imposed for the pooling layer and a dropout probability of 0.4 was imposed for the hidden layer [29].

In addition, all of the top pooling layers included rectifier linear unit (ReLU) functions and $2 \times 2$ frames. The sizes of the three fully connected networks (FCN) were 4096, 1000 and 1000. A pre-trained VGG-16 model [44] was employed, which resulted in an improved accuracy. As this was an anti-spoofing liveness detection issue with two classes (real and fake), the output of the final fully connected layer was adjusted from 2622 to 2, where 2622 was the number of facial recognition domains [45]. The Adam optimizer was utilized after increasing the learning rate to $10^{-4}$ and decreasing the weight to $10^{-6}$. The decision function was also altered from a Softmax to a sigmoid function, which is commonly used in binary classification.

*3.4. LSTM Layer*

Between the sigmoid layer and the FCN, an LSTM layer was added to learn the temporal structures of the input sequences and complete the patch-based LSTM–CNN architecture. The LSTM layer contained network nodes that were built on memory cells that could maintain their state over time and nonlinear filtering features that could control the flow of information to and from the cells, in contrast to the CNN design [46]. This architecture enabled the combination of the CNN and LSTM to recover spatial–temporal information from the frames. The total number of parameters was slightly higher than that in conventional CNN architectures, but it had a greater representation capacity.

Only one LSTM layer was employed in this approach as adding additional layers did not improve the overall performance for solving the facial anti-spoofing likeness detection issue. As the number of input pictures was fixed, the design predicted one class at each time step before employing pooling techniques to produce the results for the video classification. All of the outputs were also stacked and described using a single sigmoid layer. The characteristics were extracted from the video sequences layer by layer, including demonstrative end-to-end characteristics that ranged from basic representations to sophisticated concepts. During the period of the relevant images, the convolutional layers at the bottom could extract properties in a dense and localized fashion [32]. The fully connected layer reduced the dimensions by combining the existing representations that had been collected by the convolutional layer.

In this research, an LSTM was applied to convert the feature maps into feature sequences after obtaining the features of the segmented patches of each frame using the VGG-16-based CNN. An FCN layer was added, which had 39 neurons and Softmax nonlinearity, in between the CNN and LSTM layers. The LSTM layer provided many advantages during the process while handling large weights and huge features from the FCN, which was added after the CNN to reduce the output features [47]. Moreover, the applied LSTM layer was developed as an internal state to update the weights of the feature vector sequences of the input patches, which enhanced the performance for identifying live frames robustly. The LSTM layer was also employed to learn the temporal structures, with the sigmoid function producing the results (fake or live) using binary classification.

## 4. Results and Analysis

The outcomes of applying the suggested strategy using two different datasets are shown in Table 1. Both sets of data included a total of 50 domains, of which only 30 were used for training and the remaining 20 were used for testing. These datasets contained both printed and video versions of the attacks that were carried out. The print attacks came in a variety of forms, including warped image attacks, cut photo attacks and genuine attacks. Every video was made by either displaying a still image or a video recording of the same user for at least 9 seconds or by having a (real) client try to access a laptop while being recorded using a built-in camera.

**Table 1.** An overview of the datasets that were used in this study.

| Dataset | Domain | Images (Real/Fake) | Resolution | Print Attack | VideoReply Attack |
|---|---|---|---|---|---|
| **REPLAY ATTACK** [48] | 50 | 300/1000 | $320 \times 240$ | Available | Available |
| **CASIA-FASD** [49] | 50 | 150/400 | $640 \times 480 \times 1280 \times 7{,}201{,}920 \times 1050$ | Available | Available |

The EER (equal error rate) and HTER (half total error rate) were employed as the key metrics to evaluate the performance of our model and compare it to other current methods. Moreover, the ISO/IEC 30107-3 metrics for anti-spoofing were also adopted for our evaluation, based on true positive (TP), true negative (TN), false positive (FP) and false negative (FN) results from the following confusion metrics [50]: attack presentation classification error rate (APCER), bona fide presentation classification error rate (BPCER) and average classification error rate (ACER). EER is a metric for biometric security systems

that is used to determine the threshold value, which is defined as the standard value when the FAR (false acceptance rate) or APCER and the FRR (false rejection rate) or BPCER are the same. FAR denotes the ratio of fake images that are misclassified as real and FRR is the ratio of real images that are misclassified as fake. Equations (1)–(4) were used for further calculations:

$$APCER = FAR = \frac{FP}{TN + FP} \tag{1}$$

$$BPCER = FRR = \frac{FN}{TP + FN} \tag{2}$$

$$HTER = ACER = \frac{FAR + FRR}{2} \tag{3}$$

$$EER = FAR - FRR \tag{4}$$

Figure 2 shows a comparison of the results for the three ISO/IEC 30107 metrics (APCER, BPCER and ACER) using the REPLAY-ATTACK [51,52] and CASIA-FASD [53,54] datasets, for which our method outperformed the two state-of-the-art methods. The results indicated the improved performance of our proposed patch-based CNN–LSTM method, which produced a higher number of true positive results and fewer false negative results than the other methods using the test frames.

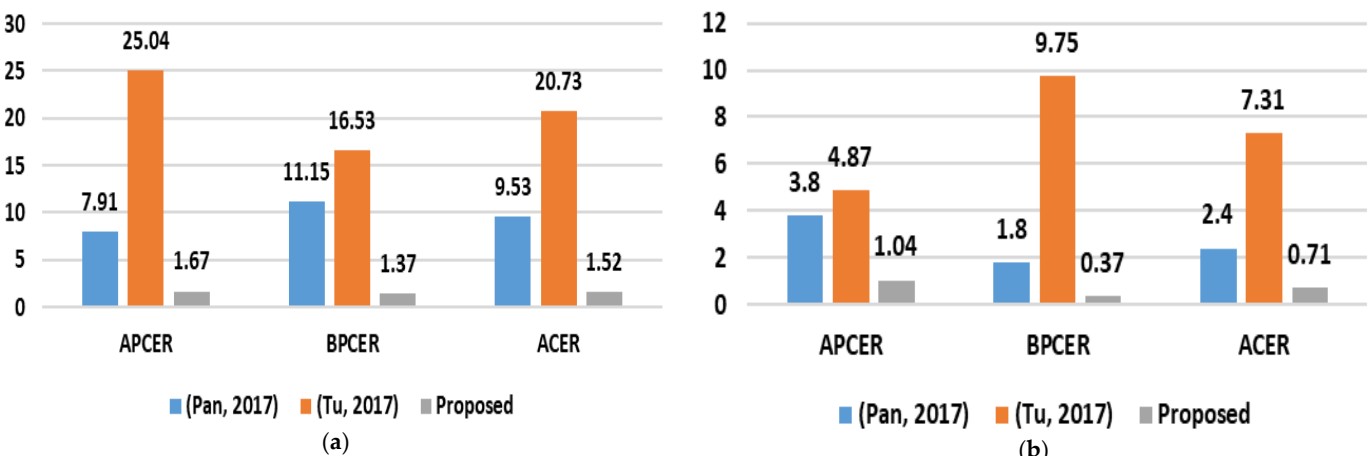

**Figure 2.** A comparison of the results from the proposed model for the ISO/IEC 30107-3 metrics using the (**a**) REPLAY-ATTACK and (**b**) CASIA-FASD datasets [55,56].

Table 2 shows our analysis results and a comparison between our method and existing approaches, based on the evaluation metrics. From the analysis, the proposed method obtained the best outcome for the CASIA-FASD dataset, producing lower HTER and EER values of 0.71% and 0.67%, respectively. However, ref. [13] produced the best result for HTER with 0% and ref. [55] demonstrated the lowest EER value of 0.10%. Although previous studies have shown better outcomes for the REPLAY-ATTACK dataset, the proposed method produced stable outcomes and maintained balanced and low HTER and EER values of 1.52% and 0.30%, respectively, which were better than most of the compared approaches.

**Table 2.** A comparison of our method to existing methods.

| Reference | Method | REPLAY-ATTACK | | CASIA-FASD | |
|---|---|---|---|---|---|
| | | HTERT | EER | HTERT | EER |
| [43] | FASNet | 1.20% | . . . . | . . . . | . . . . |
| [56] | CNN with Deep Representation | . . . . | 0.75% | . . . . | |
| [13] | Neural Network with Facial OFM Maps | 3.83% | 2.50% | . . . . | 19.81% |
| | Neural Network with Scene OFM | 3.50% | 6.16% | . . . . | 18.33% |
| | Neural Network with Multi-Cue Integration | * 0% | 0.83% | . . . . | 5.83% |
| [57] | LSTM and Rest | 1.18% | 1.03% | 1.22% | 1.00% |
| [45] | DPCNN | . . . . | 4.50% | 6.10% | 2.90% |
| [30] | Patch- and Depth-Based CNN | 0.72% | 0.79% | 2.27% | 2.67% |
| [58] | Markov Features and SVM | 4.40% | 4.00% | . . . . | 8% |
| [5] | CNN with RI-LBP | 2.60% | 2.30% | .... | 4.40% |
| [59] | CNN and SWLD | 0.69% | 0.53% | 2.14% | 2.62% |
| [60] | Dynamic Mode Decomposition with LBP and SVM | 3.70% | 5.30% | . . . . | 21.70% |
| [61] | Scale Space with LBP | 3.10% | 0.70% | . . . . | 4.20% |
| [55] | SURF and Fisher Vector Encoding | 2.20% | * 0.10% | . . . . | 2.80% |
| **Proposed Method *** | Patch-Based Modified CNN with LSTM | 1.52% | 0.30% | * 0.71% | * 0.67% |

' . . . .' = not available and '*' = highest performing outcome.

## 5. Conclusions

Facial recognition is a common form of biometric identification because it is reliable and effective. It is used for authentication purposes in a wide variety of software applications, such as automated teller machines (ATMs) and smart security systems. One of the drawbacks of this technology is that it is vulnerable to spoofing attacks, in which an imposter attempts to gain access to a system by providing the sensor with a photo of a genuine user. In this kind of attack, the impostor can pose as a legitimate user to gain access to the system. Consequently, the facial liveness detection phase of the authentication process is an essential step that must be completed before the user can be granted access.

Within the scope of this research, we developed a patch-based convolutional neural network (CNN) for facial liveness detection to improve security. This CNN was based on the VGG-16 architecture and had a deep layer. The input pictures were successively relayed to the CNN, which acted as the front-end of the architecture once the patches were produced. The CNN outputs were processed using an LSTM (long short-term memory). The LSTM recognized the temporal information in the sequences and categorized the dense layers in the neural network outputs as either authentic or fake. Two datasets (REPLAY-ATTACK and CASIA-FASD) were used to test the proposed method. According to our findings, the suggested approach produced the best performance for the CASIA-FASD dataset, with HTER and EER scores of 0.71% and 0.67%, respectively. Our proposed method also produced consistent outcomes for the REPLAY-ATTACK dataset while maintaining balanced and low HTER and EER values of 1.52% and 0.30%, respectively. Therefore, this approach could be employed in facial biometric authentication systems to identify liveness and improve authentication system efficiency.

**Author Contributions:** Conceptualization, D.A.M.; methodology, M.I.P.; software, M.I.P.; validation, M.I.P.; formal analysis, M.I.P.; investigation, M.I.P.; resources, M.I.P.; data curation, M.I.P.; writing—original draft preparation, D.A.M.; writing—review and editing, D.A.M.; visualization, D.A.M.;

supervision, S.Y.T.; project administration, S.Y.T.; funding acquisition, S.Y.T. All authors have read and agreed to the published version of the manuscript.

**Funding:** This research was funded by the Universiti Kebangsaan Malaysia (grant numbers: GGPM-2018-011 and GUP-2020-060).

**Institutional Review Board Statement:** Not applicable.

**Informed Consent Statement:** Not applicable.

**Data Availability Statement:** Not applicable.

**Conflicts of Interest:** The authors declare no conflict of interest.

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
