# Peer review of "A Patch-Based CNN Built on the VGG-16 Architecture for Real-Time Facial Liveness Detection"

_sustainability, doi:10.3390/su141610024_

Round 1

Reviewer 1 Report

This paper proposed a patch-based CNN built on the VGG-16 architecture to detect the face in the real time. Some comments are listed as follows:

1. The novelty and contribution of this paper should be demonstrated more clearly.

2. In Section 3.2, the authors seem to pay more attention to the previous research of CNN. I suggest that it is necessary to paid more attention to the design of the new method in this part.

3. On line 237, EER and HTER are presented to be the important evaluation indicators of this study, the explanation of EER should be more detailed.

4. “e” and “S” seem to be the key parts of the formulation (1), I think the selection of these two elements should be described in more detailed.

5. From Table 2, it can be seen that the new method is applied in two datasets. On CASIA-FASD dataset, it has the best recognition, but on REPLAY-ATTACK dataset shows the general performance. The Authors should use more datasets to prove the effectiveness of this method.

6. I also suggest authors should discuss some pros and cons of considered problem to clearly identify the benefits.

Author Response

We would like to thank the reviewer for your detailed comments and suggestions for the manuscript. The comments have identified important areas which required improvement. After completion of the suggested edits, the revised manuscript has benefitted from an improvement in the overall presentation and clarity. You will find a point-by-point description of how each comment was addressed in the manuscript in the file below.

Reviewer 2 Report

The paper deals with an interesting topic face analysis by the usage of deeep learining. In the present form the paper needs improvements. Starting from the introduction authors should provide a wider picture on the research domain in order to better conestualise the proposed approach. Some more references should be added as for nstance the following ones:

Moos, S. et al. (2020). Analysis of RGB-D camera technologies for supporting different facial usage scenarios. Multimedia Tools and Applications79(39), 29375-29398.

Chan, M., et al. (2005, September). Comparative study of 3d face acquisition techniques. In International Conference on Computer Analysis of Images and Patterns (pp. 740-747). Springer, Berlin, Heidelberg.

Regadring the methodological section authors should provide more details for whatc concerns the theoretical framework, in order to enphatise better the scientific added values. For what concerns the experimental validation some more lines about the experimental setting should be added, together with some more information concerning the usages scenarios, in order to have more details on the methodology performance and reliability 

Author Response

(The authors gave the same response as above.)

Reviewer 3 Report

The authors proposed an algorithm for face detection in security applications based on VGG-16 CNN with the addition of the LSTM layer. 

The paper is well structured explaining the methods from the theoretical background, leading toward experiments and evaluation at the end.

Authors should elaborate why the VGG-16 CNN structure is used and what is the advantage over other CNN structures?

Moreover, the rationale about the use of LSTM layer is not detailed enough. How this layer enables the face liveness detection on the image? How this is connected to the patch-based approach? The characteristics of the LSTM layer should be elaborated more deeply.

While results and methodology are clearly presented, it would be interesting to see a more in-depth explanation of the authors’ part of the algorithm and less explanation of standard VGG-16 architecture. 

Also, the authors didn’t present results with additional ISO metrics (APCER, BPCER, ACER). Why authors didn’t use the traditional evaluation metrics for the performance evaluation?

The authors should find and correct in the paper ... LSTM is not an abbreviation for “linear support vector machine”.

The reference list is also fine but could be better. Authors are advised therefore to check the following paper where is used the similar approach based on CNN + LSTM in similar application:

-       https://www.mdpi.com/1424-8220/22/8/3070

Additionally, authors should include in the list more recent papers because this field is under a rapid development in the last three year, for example:

- W.D. Heaven, “How to stop AI from recognizing your face in selfies,” MIT Technology Review, May 5, 2021

- K. Suh, Kun, E.C. Lee, “Face liveness detection for face recognition based on cardiac features of skin color image,” 100110C. 369 10.1117/12.2242472

- Z. Ming, M. Visani, M. Luqman, J.C. Burie, “A survey on anti-spoofing methods for face recognition with RGB cameras of 371 generic consumer devices,” Cornell University, 2020

- etc.

The reference list should be complete including DOI numbers and other relevant information where possible.

Nevertheless, the authors present a consistent paper with relevant references and consistent conclusions. 

Author Response

(The authors gave the same response as above.)

Round 2

Reviewer 2 Report

Paper has been adequately improved.

Reviewer 3 Report

Authors have properly enriched their work, by addressing each comment in a suitable way. The paper turns out to be notably improved.